# Assessment of time management practice and associated factors among primary hospitals employees in north Gondar, northwest Ethiopia

Muluken Genetu Chanie[1], Erkihun Tadesse Amsalu[2]*, Gojjam Eshete Ewunetie[3]

**1** School of Public Health, Department of Epidemiology and Biostatistics, College of Medical and Health Sciences, Wollo University, Dessie, Ethiopia, **2** Department of Epidemiology and Biostatistics, School of Public Health, College of Medical and Health Sciences, Wollo University, Dessie, Ethiopia, **3** Dembia General Hospital, Outpatient Department, Gondar, Ethiopia

* erkihunt@yahoo.com

**Data Availability Statement:** All relevant data are within the manuscript and its Supporting Information files.

## Abstract

### Background

Time management practice can facilitate productivity and success, contributing to work effectiveness, maintaining balance and job satisfaction. Thus, this study aimed to assess time management practices and associated factors among employees of primary hospitals in north Gondar.

### Methods

An Institutional based cross-sectional study among primary hospital employees in north Gondar was conducted from March to April 2018. A structured and pre-tested questionnaire was used to collect the data. Simple random sampling technique was utilized to select 422 employees. Bivariate and multivariate logistic regression model were done to identify factors associated with time management practice. Adjusted odds ratio (AOR) with a 95% confidence interval (CI) was ascertained to show the strength and direction of association.

### Result

In this study, the prevalence of time management practice was 56.4% (95%CI: 49.3, 61.7). Being satisfied with organizational policies (AOR = 2.16; 95%CI: 1.02–4.68), performance appraisals (AOR: 2.11; 95%CI: 1.32–4.66), compensation and benefits (AOR: 4.18; 95%CI: 2.18–7.99), and planning (AOR: 2.86; 95% CI: 1.42–5.75) were statistically significant factors associated with time management practice.

### Conclusion and recommendation

The overall time management practice among the primary hospital employees was low. Planning, organizational policy, compensation and benefit, performance appraisal, and residence were factors significantly associated with hospital employee's time management

**Funding:** The authors received no specific funding for this work.

**Competing interests:** The authors have declared that no competing interests exist.

**Abbreviations:** ANRHB, Amara National Regional Health Bureau; AOR, Adjusted Odds Ratio; COR, Crude odds ratio; CI, Confidence Interval; EPi-Info, Epidemiological Information; ETB, Ethiopian Birr; UAE, United Arab Emirates; USD, United States Dollar.

practice. Thus managers and employees need to carry out interventions on significant factors to improve the employees' time management practice.

## Introduction

Time is described as a measure of the duration and order of events in the past, the present and the future [1]. Time is a priceless resource and continues to pass by without coming back. Time itself cannot be managed because it is an inaccessible factor rather task in line with time [2]. The concept of time management started with industrial revolution and became the modern notion of doing things effectively and efficiently [3, 4]. Hence its importance has been increasing from day to day [5]. Effects of time management are a panacea to organizational effectiveness. It is difficult to measure time management practice but largely depends on the outcomes of employees performance [6].

Time management practice is the act of influencing one's key behavioral dimensions to complete as many tasks as possible within a given time period. Such behavioral dimensions include organization of work and continuous application of time management techniques as a habit [7].

According to Donaldson, "the aim of good time management is to achieve the lifestyle balance you want". Good time at work means doing high quality work, not high quantity [8]. Good time management such as setting goals and priorities as well as scheduling and delegation of tasks can facilitate productivity and success, contributing to work effectiveness, maintaining balance and job satisfaction [9]. In contrast, poor time management has been associated with poor work quality, low productivity, negative influence on career path, and high stress levels [10,11,12].

A study has pointed out that time management practices in different countries of the world vary. A study conducted on factors affecting time management and nurses' performance in Hebron hospital, Palestine, reported time management practice was high with rate 69.5% [13]. A study conducted in Pakistan on time management and organizational performance in 2011 revealed that among 1200 participants overall time management practice was 30% [14]. A study done in UAE(United Arab Emirates) revealed that time management practice was 49% and the study showed that 56% of employees lack planning, prioritizing and scheduling time for their work [15].

A study done in Egypt on time management program showed that time management practice was 45% among the studied head nurses at their work site with impacts on their job performances and performance appraisals [16].

Another study done in Nigeria on time management in Nigerian hospitals showed that time management practice among employees was found 51% [17].

A study done in Ethiopia on the effect of time management practice among Dire-Dawa university students showed that time management practice was found 34% [18].

There is prevalent lack of time management culture in many societies especially in developing countries including Africa which may be detrimental to both the organization as well as the employees [17]. Most people feel like they have too much to do and not enough time and they blame lack of time for their unachieved goals, poor performance and low productivity [14].

There were several factors known to contribute to poor time management practices. Among these factors effective time management method related factors, personal factors (punctuality, time wasters), administrative and organizational obstacles of time management (organizational policy, Lack of incentives, performance appraisal), and employees performance in an organization are the most important factors which have important role in determining employees' time

management practices [13, 19, 20]. Other variables such as education, age, marital status, and sex also has been determined as factors that contribute to it [19, 21].

There was an implementation of time management practices in every organization but still there is a gap in reaching productivity and improved performance of employees.

There were limited previous studies regarding time management practice, but there were no published data about time management practice among primary hospital employees in Ethiopia including the study setting.

Therefore, this study aimed to assess time management practices and associated factors among employees of primary hospitals in North Gondar Zone.

## Method and materials

### Study setting

The study was conducted at primary hospitals found in North Gondar Zone, which is one of the 11 zones found in the Amhara National Regional State. This zone is located in Northwest part of Ethiopia and divided into 24 woredas (Districts). There are 9 primary hospitals, 126 health centers and 573 health posts in this zone. In all 9 hospitals there are about 1,071 employees of which 543 were health professionals and 526 were supportive workers according to North Gondar health department's plan office data.

### Study design and period

Institutional based cross-sectional study was conducted from March to April 28, 2018.

### Population and sample

Debark hospital, Metema hospital; Delgi hospital, Gohala hospital, Aykel hospital, Mekanebrhan hospital, Amba-giorgis hospital, Koladiba hospital, and Abraha–Jira hospital were selected hospitals from Gondar zone for this study. There were a total of 1,071 employees working in the selected primary hospitals of north Gondar zone. All employees working in primary hospitals of north Gondar zone for at least 6 months period were included into the study. Those employees who had worked for less than six months in the hospital and respondents with incomplete data were excluded from the study. A total of 391 employees which fulfilled the inclusion criteria were participated in the study.

### Sample size determination and sampling procedure

Sample size for the prevalence part of the study was determined by using single population proportion (as there were no previous study conducted in the area) formula considering the following assumptions: taking 50% prevalence of time management practices and expected margin of error (d) 0.05 and with 95% confidence level (Z $_{a/2}$) $n = \frac{(z_{\alpha/2})^2 p(1-p)}{d^2} = \frac{(1.96)^2 0.5(0.5)}{(0.05)^2} = 384$

By adding 10% for non-respondents the total sample size was 422.

For associated factors of time management practice the sample size was determined by using double population proportion formula using selected three key predictors [22,23] according to the following assumptions and computed by Epi-info version 7 software (Table 1).

Thus the minimum adequate sample size for this study was 422 taken from single population proportion formula.

Proportional allocation of 422 samples was done for each primary hospital based on the number of employees working in the respective hospitals. Then study participants were

**Table 1. Sample size determination for factors associated with time management practice among primary hospital employees.**

| No | Associated factors | Assumptions | Final sample size |
|---|---|---|---|
| 1 | Planning | OR = 1.995, ratio 1:1, planning in unexposed group = 47%,power = 80%, at 95% confidence level and 10% for non-response rate | 323 |
| 2 | Time wasters | OR = 2.067, ratio 1:1, time wasters in unexposed group = 44.5%, power = 80%, at 95% confidence level and 10% for non-response rate | 293 |
| 3 | Procrastination | OR = 2.589, ratio 1:1, procrastination in unexposed group = 43.3%, power = 80%, at 95% confidence level and 10% for non-response rate | 178 |

selected by simple random sampling method in each Hospital. A total of 84 respondents from Debark hospital, 87 from Metema hospital, 36 from Delgi hospital, 28 from Gohala hospital, 44 from Aykel hospital, 29 from Mekanbrhan hospital, 41 from ambagiorgis hospital, 45 from Koladba hospital, and 28 from abraha-jira hospital were selected.

## Study variables and data collection procedure

Time management practice was used as a dependent variable. Socio-demographic factors (sex, age, residence, marital status, educational status, type of profession, and work experience), personal factors (time wasters, procrastination, and punctuality), administrative and organizational factors (organizational policy and strategy, work environment, compensation and benefit, performance appraisal, recognition, and promotion), and employees performance (planning, implementation, and responsibility) were independent variables of the study.

Time management practice was defined as scheduled use of time by employees at work site. It is measured by 5 items each scored a 5 point Likert-scale with 1 denoting strongly disagrees and 5 denoting strongly agree. After dichotomous category, responses above and equal to 65% was categorized as good time management practices [13].

Organization policy and strategy was described as the respondent's feeling on the application of organizational policies and strategies. It was measured by using 3 items each scored 5-point Likert scale. It was categorized as satisfied if the responses were ≥ the mean score value and unsatisfied if the responses were below the mean score value.

Responsibility was described as the respondent's duty to fulfill a responsibility as personal and as member of team work. It is measured by using 3 items each scored 5-point Likert scale. It was categorized as high if the responses were ≥ the mean score value and low if the responses were below the mean score value.

Work Environment was described as the quality of the working environment both its physical attributes and the degree to which it provided meaningful work condition. It was measured by using 5 items each scored 5-point Likert scale. It was categorized as good if the responses were ≥ the mean score value and bad if the responses were below the mean score value.

Compensation and benefit was described as employees feeling of fairness and adequate payment for work done and financial rewards for better performance. It was measured by using 3 items each scored 5-point Likert scale. It was categorized as satisfied if the responses were ≥ the mean score value and unsatisfied if the responses were below the mean score value.

Recognition and Promotion was described as employees feeling of recognition and promotion systems of the hospital. It was measured by using 4 items each scored 5-point Likert scale. It was grouped as satisfied if the responses were ≥ the mean score value and unsatisfied if the responses were below the mean score value.

Performance appraisal was described as the participants feeling on measurement of their actual performance. It was measured by using 3 items each scored 5-point Likert scale. It was

categorized as satisfied if the responses were $\geq$ the mean score value and unsatisfied if the responses were below the mean score value.

Procrastination was described as the employees postponing of scheduled tasks. It was measured by using 4 items each scored 5-point Likert scale. It was categorized as high if the responses were $\geq$ the mean score value and low if the responses were below the mean score value.

Time wasters were described as the engagement to an activity that spends employees' productive time at work site. It was measured by using 5 items each scored 5-point Likert scale. It was categorized as high if the responses were $\geq$ the mean score value and low if the responses were below the mean score value.

For this study the data was collected by using self-administered structured questionnaires adopted from advanced corporate training and legal management consultants [24, 25]. Three data collectors (diploma nurses) and two supervisors (BSc Nurses) were recruited. One day training was given by the principal investigator for data collectors about the objectives and processes of data collection. Pre-test was conducted on 10% of total sample size (42 employees) at Addis zemen primary hospital. All filled questionnaires were checked by the principal investigator for its completeness and consistency.

## Data management and analysis

Prior to the actual data collection, frequent supervision was done, interviewers were trained, and interviews were performed using the local language Amharic. Reliability test (Cronbach's alpha) was performed to check reliability of the questionnaire items. Data were checked for completeness, organized and entered into Epi-info version 7, and then exported to STATA version 14 software for analysis.

Tables were used to present the results. Descriptive statistical analysis such as frequencies and percentages were used to describe the characteristics of the study population. As the response variable i.e. time management practice was dichotomous (poor, good), logistic regression was used to identify factors that affect time management practice. Variables with $\leq$ 0.05 p-values in the bi-variable analysis were fitted in the multivariable model. Adjusted Odds Ratio (AOR) with a 95% Confidence Interval (CI) and p-value <0.05 in the multivariable model were used to declare significant association with time management practice. Goodness of fit was checked using Hosmer Lemeshow test (p = 0.187).

## Ethical consideration

Ethical clearance was obtained from the University of Gondar Ethical Review Board (IRB). Before communicating study participants' official permission letter of cooperation was obtained from Amara National Regional Health Bureau (ANRHB). The purposes and the importance of the study were explained and informed consent was secured from each participant. Respondents were clearly told about the study and the variety of information needed for them. They were given the chance to raise any question about the study and free to refuse or terminate the interview at any moment. Name of participants and any personal identifiers were not included in the study, and the confidentiality of the data was kept at all level of the study.

## Results

### Socio -demographic characteristics of respondents

From a total of 422 primary hospital employees, 391 of them went into the analysis. The remaining 31 were excluded from the study due to incomplete information. More than half, 232 (59.3%) were males and 234(59.8), highlanders. Regarding educational status, 254 (65%)

of the respondents had diploma and below. The median age of respondents were 28.5 years (IQR: 25.82–31.25). Regarding salary, 194(49.6%) of employees earned less than 3137 Ethiopian Birr (ETB) i.e. below 109 USD monthly (**Table 2**).

## Organizational related factors of respondents

In this study majority of employees about 243(62.1%) were unsatisfied with organizational policy. similarly about 239(61.1%) of respondents were unsatisfied with performance appraisal. Regarding compensation and benefit about 348 (89.0%) were unsatisfied. About the working environment, 119 (30.4%) of employees had good Work environment. The finding also showed that about 296 (75.5%) of respondents were unsatisfied with recognition and reward (**Table 3**).

## Employee performance related factors

This study showed that majority of primary hospital employees, 316 (80.8%) had planning for their work. More than half, about 222 (56.8%) of employees showed high responsibility for their work. Regarding implementation about 307(78.5%) had high implementation for their work (**Table 3**).

## Personal related factors of respondents

Regarding punctuality the majority of respondents about, 337(86.2%) was punctual for their work. Similarly majority of employees about, 317(81.1%) had high procrastination. This study also showed that majority of respondents about, 335 (85.7%) were high time waster (**Table 3**).

The overall prevalence of time management practice among primary hospital employees was 56.4% (95%CI: 49.3%, 61.7%).

**Table 2. Socio-demographic characteristics of primary hospital employees in north Gondar zone, 2018 (n = 391).**

| Variable | Category | Frequency | Percentage |
|---|---|---|---|
| Age (years) | 20–24 | 55 | 14.1 |
| | 25–29 | 178 | 45.5 |
| | 30–34 | 107 | 27.4 |
| | ≥35 | 51 | 13.0 |
| Sex | Male | 232 | 59.3 |
| | female | 159 | 40.7 |
| Marital status | Single | 194 | 49.6 |
| | married | 179 | 45.8 |
| | divorced | 18 | 4.6 |
| Educational level | diploma and below | 254 | 65.0 |
| | degree | 127 | 32.5 |
| | masters | 10 | 2.5 |
| Religion | orthodox | 377 | 96.4 |
| | Muslim | 11 | 2.8 |
| | protestant | 3 | 0.8 |
| Residence | lowland | 157 | 40.2 |
| | highland | 234 | 59.8 |
| Salary(ETB) | <3137 | 194 | 49.6 |
| | 3137–4086 | 81 | 20.7 |
| | 4086–4726 | 41 | 10.5 |
| | >4726 | 75 | 19.2 |

**Table 3. Organizational policy, employee's performance and personal related factors of respondents in North Gondar zone, 2018 (n = 391).**

| Variable | Category | Frequency | Percentage |
|---|---|---|---|
| Punctuality | Yes | 337 | 86.2 |
| | No | 54 | 13.8 |
| Organizational policy | Satisfied | 148 | 37.9 |
| | unsatisfied | 243 | 62.1 |
| Performance appraisal | Satisfied | 152 | 38.9 |
| | unsatisfied | 239 | 61.1 |
| Work environment | Good | 119 | 30.4 |
| | bad | 272 | 69.6 |
| Compensation and benefit | satisfied | 43 | 11.0 |
| | unsatisfied | 348 | 89.0 |
| Recognition and reward | satisfied | 95 | 24.3 |
| | unsatisfied | 296 | 75.7 |
| Planning | Yes | 316 | 80.8 |
| | No | 75 | 19.2 |
| Implementation | High | 307 | 78.5 |
| | Low | 84 | 21.5 |
| Responsibility | High | 222 | 56.8 |
| | Low | 169 | 43.2 |
| Procrastination | High | 317 | 81.1 |
| | Low | 74 | 18.9 |
| Time waster | High | 335 | 85.7 |
| | Low | 56 | 14.3 |

## Factors associated with time management practice

In the bi-variable analysis punctuality, organizational policy, performance appraisal, work environment, recognition and reward, planning, implementation, compensation and benefit, residence, procrastination, and time waste level were factors found to besignificantly associated with time management practice at p- value$\leq$0.05.

However, in the multivariable mixed effect logistic regression analysis, organizational policies, planning, performance appraisal, compensation and benefit, and residence were factors significantly associated with time management practice among primary hospital employees.

Hospital employees who were satisfied with organizational policies and strategies were nearly two times (AOR = 2.16, 95% CI: 1.021, 4.69) more likely to have good time management practice compared with unsatisfied employees.

Employees who were satisfied with performance appraisals were two times (AOR = 2.11, 95% CI: 1.32, 4.67) more likely to have good time management practice compared with unsatisfied counterparts.

Similarly employees satisfied with compensation and benefit were nearly four times (AOR = 4.18, 95% CI: 2.19, 7.99) more likely to have good time management practice compared with unsatisfied employees.

On the other hand employees who were good in planning were nearly three times (AOR = 2.86, 95% CI: 1.42, 5.75) more likely to have good time management practice compared with those poor in planning.

Employees working in the highland areas were nearly two times (AOR = 2.08, 95% CI: 1.08, 4.01) more likely to have good time management practice compared with those working in lowland (**Table 4**).

## Discussion

In this study, the prevalence of time management practice among employees was low. This finding was higher as compared to studies conducted in Nigeria on time management practice 51% [17], and a study done on time management program on job satisfaction in Egypt 45% [16]. Similarly, this finding was much higher than studies conducted in United Arab Emirates which was 49% [15] and Pakistan which was reported as 30% [14]. The finding was also higher than a study done in Dire-Dawa University Ethiopia which reported the prevalence of time management practice was 34% [18].

However, it is lower than studies conducted in Palestine on health professionals performance in Hebron Hospital, in which the prevalence of time management practice was reported 69.5% [13] and in Australia on health professionals the prevalence was reported 64% [26]. This discrepancy could be resulted from differences in infrastructure in the health institutions, study setting differences, and differences in the respondents which could affect the status of time management practice. In this study the study subjects were primary hospital employees working in public hospitals only, where as in Palestine and Australia the study subjects were health professionals working both in selected public and private hospitals [13, 26].

This study identified hospital employee's time management practices were influenced more by organizational policy and strategy. Hospital employees who were satisfied with

**Table 4. Bi-variable and Multivariable logistic regression analysis of factors associated with time management practices among primary hospitals employees in north Gondar zone from March to April 28, 2018 (n = 391).**

| Variable | Category | Time management practice | | COR(95% CI) | AOR(95%CI) |
|---|---|---|---|---|---|
| | | good | Poor | | |
| Punctuality | Yes | 39 | 15 | 0.63(0.33–1.22) | 0.67(0.28–1.65) |
| | No | 271 | 66 | 1 | 1 |
| Organizational policy | satisfied | 132 | 16 | 3.01(1.67–5.44) | 2.16(1.02–4.68)* |
| | unsatisfied | 178 | 68 | 1 | 1 |
| Performance appraisal | satisfied | 134 | 18 | 2.67(1.51–4.71) | 2.11(1.32–4.67)* |
| | unsatisfied | 176 | 63 | 1 | 1 |
| Work environment | good | 96 | 23 | 1.13(0.66,0.94) | 0.78(0.39–1.57) |
| | bad | 214 | 58 | 1 | 1 |
| Recognition and reward | satisfied | 84 | 11 | 2.36(1.19–4.68) | 1.53(0.65–3.57) |
| | unsatisfied | 226 | 70 | 1 | 1 |
| Planning | yes | 264 | 52 | 3.20(1.84–5.56) | 2.86(1.42–5.75)** |
| | no | 46 | 29 | 1 | 1 |
| Implementation | high | 259 | 48 | 3.49(2.04–5.96) | 1.87(0.91–3.88) |
| | low | 51 | 33 | 1 | 1 |
| Compensation and benefit | satisfied | 199 | 23 | 4.52(2.65–7.73) | 4.18(2.19–7.99)** |
| | unsatisfied | 111 | 58 | 1 | 1 |
| Residence | lowland | 178 | 56 | 0.60(0.36–1.02) | 2.08(1.08–4.01)* |
| | highland | 136 | 25 | 1 | 1 |
| Procrastination | high | 259 | 23 | 2.01(1.14–3.56) | 1.33(0.06–2.83) |
| | low | 51 | 66 | 1 | 1 |
| Time waster | high | 269 | 66 | 1.49(0.78–2.85) | 0.96(0.39–2.33) |
| | low | 41 | 15 | 1 | 1 |

COR: Crude odds ratio, CI: Confidence interval, AOR: adjusted odds ratio, 1: Reference category

*: significant at p< 0.05

**: Significant at p< 0.001

organizational policies and strategies had good time management practice. This finding is consistent with studies conducted in Pakistan and Palestine [13, 14[. The possible explanation could be if employees perceived that the policies and strategies of the organization are not fair, they become disappointed and dissatisfied which could affect employee's time management practice negatively. Whereas satisfied respondents would be motivated and can manage their time effectively in the organization because there is fairness in, and benefits from their organizations [14, 27, 28].

Time management practice was high among the respondents who were satisfied with performance appraisal got from the hospital when compared with unsatisfied employees. This finding is supported by studies done on time management [20]. This might be due to unfair relationships between some workers and with the department head and/or the hospital manager who may treat some staffs better than others based on some form of personal relationships [20, 29].

Another strong significant predictor of time management practice from hospital employees was compensation and benefit. Primary hospital employees who were satisfied with compensation and benefit had good time management practice when compared with unsatisfied. The result of this finding is in line with studies conducted in Egypt about the effect of time management program on job satisfaction [16]. This could be explained by poor working environment, dissatisfaction with the organization, less professional opportunity because it does not give them a chance to grow and develop their own abilities, which in turn lead to poor time management practices of employees[29,30].

Primary hospital employees who were good in planning were good in time management practice. This finding is in line with a study done on time management and academic performance in United Arab Emirates [15]. The possible explanation could be good experience in planning decreases employees dependence on others what to do and task leading activities timely [27, 31, 32].

With respect to socio-demographic characteristics, residence was significantly associated with hospital employee's time management practice. Employees working in the highland areas were more likely to manage their time when compared with employees working in lowland. This finding is consistent with studies done on a cross-cultural investigation of time management practices and job outcomes [33]. This finding could be explained by the fact that differences in working settings and weather conditions respondents were working [32, 33].

The literature is not consistent in terms of the relationship between some associated factors and employees' time management practice. In this particular study time management practice factors like punctuality, work environment, procrastination, time waster, recognition and reward, implementation, marital status and work experience were not found to be statistically significant associations with employees' time management practice. This might be due to differences in the content and nature of work in the hospitals and difference in research settings might cause discrepancy results.

The variables used in the study might not be exhaustive and some other variables might be missed that need to be tested for association with time management practice. Use of self-reporting measures may have some potential for reporting biases, which may have occurred because of the respondents' interpretation of the questions; they may over or under report a phenomenon. Finally, the study lacked the component of follow-up, in which the researcher could compare participants' time practice versus their actual practice.

The study could have valuable implication to formulate appropriate strategies by different stakeholders involved in hospital administration and management programs, and other related public health interventions. It is also important to enhance and upgrade participants' understanding about time management; and to equip them with major techniques of time

management. This study could also provide information to subsequent researchers on time management practices and factors associated with it.

## Conclusion

The overall time management practice among the primary hospital employees was low. Planning, organizational policy and strategy, compensation and benefit, performance appraisal, and residence were predictors significantly and positively associated with hospital employee's time management practice.

Thus it is recommended that hospital employees must give special emphasis about planning to improve their time management practice. Hospital Managers need to assess and improve organizational policies and strategies and performance appraisal systems practicing in the respective hospitals to treat employees fairly and equally. It is also recommended to conduct further research on the issue by extending the study setting and the study population.

## Supporting information

**S1 File. Questionnaires sheet.**
(DOCX)

## Acknowledgments

First we would like to thank all study participants for their cooperation in providing the necessary information. We would also thank data collectors and supervisors for the devotion and quality work during data collection for the accomplishment of this work.

## Author Contributions

**Conceptualization:** Muluken Genetu Chanie, Erkihun Tadesse Amsalu.

**Data curation:** Erkihun Tadesse Amsalu, Gojjam Eshete Ewunetie.

**Formal analysis:** Muluken Genetu Chanie, Erkihun Tadesse Amsalu.

**Investigation:** Muluken Genetu Chanie.

**Methodology:** Muluken Genetu Chanie.

**Visualization:** Erkihun Tadesse Amsalu.

**Writing – original draft:** Erkihun Tadesse Amsalu.

**Writing – review & editing:** Muluken Genetu Chanie, Erkihun Tadesse Amsalu, Gojjam Eshete Ewunetie.

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
