## [Decision Letter · Decision Letter 0]

25 Sep 2019

PONE-D-19-24146

Assessment of time management practices and associated factors among employees of primary hospitals in north Gondar, northwest Ethiopia

PLOS ONE

Dear Dr Amsalu,

Thank you for submitting your manuscript to PLOS ONE. After careful consideration, we feel that it has merit but does not fully meet PLOS ONE’s publication criteria as it currently stands. Therefore, we invite you to submit a revised version of the manuscript that addresses the points raised during the review process.

I acclaim the authors for taking their time to study this very sensitive topic especially in Ethiopia set up.

Abstract: briefly articulated and has clearly shown the overall study.

**Introduction:**

This section has tried to explain time and time management practice at a different perspective. It looks good and explanatory. It would be very good if you quote studies in Africa that could deliver better information about time management. Perhaps, that would give a good picture, avoid bias and pointless conclusion. Similarly, a study conducted on Africans resided in Iran, as mentioned in this study, which might have different social fabrics when compared to studies done in other African countries, could be susceptible to bias. Hence, I recommend using studies that have been done in Africa or at least in a similar setting to offer conceivable information.

As a whole, the literature reviews looks very scanty and insufficient to demonstrate time management and practice especially in Ethiopia settings. Perhaps, it would be very useful to incorporate studies that could at least be related to the study setting to make the study more useful and relevant.

**Methodology:**

In general the methodology appears so murky and narrow to provide enough information. This section needs outright revision or overhaul. it has short of scientific explanation, doesn’t appropriately depict methods and has overlooked key research tools. i found the following point unclear and need to be addressed.

How did you determine the sample size? How did you allocate study participants? How did you select the study participant? Where did you get or how did you develop data collection tools or questionnaire in this study? How did you measure the validity and reliability of the study tools or questionnaire implemented to collect data in this study?  It is not also clear how the cutoff point made for the Likert scale.  How the questionnaire contents organized and formulate to address the objective?  

What is the dependent variable in this study? It looks like level of time management practice appears to be the dependent variable. However, you have operationally defined time management practice on the next paragraph. You should be clear otherwise it would be very confusing.  It looks like there is misconception between time management practice and level of time management practice in this study. Hence, you need to differentiate or clarify both first, then clearly address dependent variable and provide the proper operational definition. Or else, this would affect the full analysis and study. Similarly, would be good to define those factors indicated in this study as well.

**Result**:

The socio-demography well stated. Maybe, it would be useful if you depict the response rate in percent than numbers for simplicity.

However, the remaining part of the result seems disorganized and superficial. It lacks coherence. It is so difficult to assert the authenticity of the analysis. It looks like there is a kind of mingling between the sub-topics “time management practices, organizational policies, employees’ performance and personal factors” and “factors associated with time management practice”. Either you need to modify the first sub topic or merge with the second topic. This section as a whole needs stringent revision.

**Discussion:**

In general, this section has tried to compare studies that has been conducted in different places and has made arguments, which is very commendable. However, the arguments are weak, frail and lack of scientific reasoning. It is not clear also, for instance, on the first paragraph you indicated that “***the proportion of time management practice among employees was 56.4%” ***what does the word proportion designate in this sentences? There is similar inconsistency throughout the document that has to be fixed.

In addition to this, you need to use studies that are closely related to your study setting at various measuring scale to make plausible comparison. Otherwise, citing studies which never have related to your study setting will affect the analysis and the entire result one or another way.

What is the limitation of your study?

**Conclusion:** looks good but what are your recommendations? You have inscribed your suggestion and recommendation at the abstract section, but not in this section.

Make sure you stick with reference regulation of the publisher.

We would appreciate receiving your revised manuscript by Nov 09 2019 11:59PM. To enhance the reproducibility of your results, we recommend that if applicable you deposit your laboratory protocols in protocols.io, where a protocol can be assigned its own identifier (DOI) such that it can be cited independently in the future. For instructions see: http://journals.plos.org/plosone/s/submission-guidelines#loc-laboratory-protocols

We look forward to receiving your revised manuscript.

Kind regards,

Solomon Assefa Woreta

Academic Editor

PLOS ONE

Journal Requirements:

Additional Editor Comments (if provided):

I acclaim the authors for taking their time to study this very sensitive topic especially in Ethiopia set up.

Abstract: briefly articulated and has clearly shown the overall study.

Introduction:

This section has tried to explain time and time management practice at a different perspective. It looks good and explanatory. It would be very good if you quote studies in Africa that could deliver better information about time management. Perhaps, that would give a good picture, avoid bias and pointless conclusion. Similarly, a study conducted on Africans resided in Iran, as mentioned in this study, which might have different social fabrics when compared to studies done in other African countries, could be susceptible to bias. Hence, I recommend using studies that have been done in Africa or at least in a similar setting to offer conceivable information.

As a whole, the literature reviews looks very scanty and insufficient to demonstrate time management and practice especially in Ethiopia settings. Perhaps, it would be very useful to incorporate studies that could at least be related to the study setting to make the study more useful and relevant.

Methodology:

In general the methodology appears so murky and narrow to provide enough information. This section needs outright revision or overhaul. it has short of scientific explanation, doesn’t appropriately depict methods and has overlooked key research tools. i found the following point unclear and need to be addressed.

How did you determine the sample size? How did you allocate study participants? How did you select the study participant? Where did you get or how did you develop data collection tools or questionnaire in this study? How did you measure the validity and reliability of the study tools or questionnaire implemented to collect data in this study? It is not also clear how the cutoff point made for the Likert scale. How the questionnaire contents organized and formulate to address the objective?

What is the dependent variable in this study? It looks like level of time management practice appears to be the dependent variable. However, you have operationally defined time management practice on the next paragraph. You should be clear otherwise it would be very confusing. It looks like there is misconception between time management practice and level of time management practice in this study. Hence, you need to differentiate or clarify both first, then clearly address dependent variable and provide the proper operational definition. Or else, this would affect the full analysis and study. Similarly, would be good to define those factors indicated in this study as well.

Result:

The socio-demography well stated. Maybe, it would be useful if you depict the response rate in percent than numbers for simplicity.

However, the remaining part of the result seems disorganized and superficial. It lacks coherence. It is so difficult to assert the authenticity of the analysis. It looks like there is a kind of mingling between the sub-topics “time management practices, organizational policies, employees’ performance and personal factors” and “factors associated with time management practice”. Either you need to modify the first sub topic or merge with the second topic. This section as a whole needs stringent revision.

Discussion:

In general, this section has tried to compare studies that has been conducted in different places and has made arguments, which is very commendable. However, the arguments are weak, frail and lack of scientific reasoning. It is not clear also, for instance, on the first paragraph you indicated that “the proportion of time management practice among employees was 56.4%” what does the word proportion designate in this sentences? There is similar inconsistency throughout the document that has to be fixed.

In addition to this, you need to use studies that are closely related to your study setting at various measuring scale to make plausible comparison. Otherwise, citing studies which never have related to your study setting will affect the analysis and the entire result one or another way.

What is the limitation of your study?

Conclusion: looks good but what are your recommendations? You have inscribed your suggestion and recommendation at the abstract section, but not in this section.

Make sure you stick with reference regulation of the publisher.

Reviewers' comments:

Reviewer's Responses to Questions

**Comments to the Author**

1. Is the manuscript technically sound, and do the data support the conclusions?

Reviewer #1: Yes

2. Has the statistical analysis been performed appropriately and rigorously? 

Reviewer #1: Yes

3. Have the authors made all data underlying the findings in their manuscript fully available?

Reviewer #1: Yes

4. Is the manuscript presented in an intelligible fashion and written in standard English?

Reviewer #1: No

5. Review Comments to the Author

Reviewer #1: Dear PLOS One thank you for the chance given to review a research article titled “Assessment of time management practices and associated factors among employees of primary hospitals in north Gondar, northwest Ethiopia”. Effective time management particular for those who are directly involved in the process of care of human being. Hence, this research will discover the colossal time management problem in the health sector. The following are my comments:

General Comments

Is there an African / Ethiopia perspective of time?

What are the dimensions of time?

Specific Comments

On the abstract Section

The objective is missed.

Avoid use of ‘predictors’ in cross sectional studies.

In key words include Gondar.

On the Introduction Section

Is this definition of time the contemporary definition?

What is special about time management for health care workers?

What are the adverse consequences of ineffective time management practice?

The flow lacks coherence. Global, regional and national data on the matter of interest were missed.

On the Methods Section

Start with study setting and tell us your reference.

You didn’t mentioned study design.

Why you used systematic sampling?

Is there no difference between male and female health care professionals? Between diploma and degree holders? Between nurses and medical doctors?

On the Result section

Why don’t you classify residence as urban and rural rather than presenting it as highlanders and lowlanders?

When do we say an individual is satisfied with performance appraisal?

What is your case to variable ratio?

Present some of your findings e.g Proportion of Procrastination

On the Discussion Section

Avoid presenting frequencies.

Reference for your justification is needed. For example “This might be due to unfair relationships between some workers and with the department..” what does that mean? Do you have data on it?

What does organizational policy? Is that norm or some other guiding document? Is that endorsed from ministry of health or?

What are the types of compensations and benefits ? do you have evidence? Needs reference?

On the Conclusion Section

You collected data of 65% from diploma holders and you are concluding for all types of health care workers?

The word ‘poor’ is not ethical.

Implications were missed

Lacks recommendation

Thank You!

6. PLOS authors have the option to publish the peer review history of their article (what does this mean?). If published, this will include your full peer review and any attached files.

Reviewer #1: Yes: Yes

---

## [Author Response · Author response to Decision Letter 0]

9 Nov 2019

Dear all,

We would like to thanks for these constructive, building and improvable comments on this manuscript that would improve substance and content of the manuscript. We considered each comments and clarification questions of editors and reviewers on the manuscript thoroughly. Our point-by-point responses for each comment and questions are described in detailed on the following pages. Further, the details of changes were shown by track changes in the supplementary document attached.

Editor questions/comments Response

Introduction

 As a whole, the literature reviews looks very scanty and insufficient to demonstrate time management and practice especially in Ethiopia settings. Perhaps, it would be very useful to incorporate studies that could at least be related to the study setting to make the study more useful and relevant.

 Thanks editor for your constructive comments. Even though there were inadequate literatures related with the problem particularly in Ethiopia we have searched effort fully and included them in the main document. 

Methodology

How did you determine the sample size? Dear editor thanks for your constructive comments. The sample size was determined using single population proportion formula, taking 50% prevalence of time management practices (as there was no previous study) with the following assumption: 95% CI and 5% margin of error.

Where 

 = 

By adding 10% for non-respondents the total sample size was 422. 

For associated factors of time management practice the sample size was determined by using double population proportion formula and computed by Epi-info version7 software. Thus the minimum adequate sample size for this study was 422 taken from single population proportion formula.

Details about sample size determination were put in the main document.

 How did you allocate study participants? 

 Thanks again for your constructive comments. Proportional allocation of study participants (422 employees) was done for each primary hospital based on the number of employees working in the respective hospitals. Details were described in the main document.

How did you select the study participant? Thanks editor for valuable comment. Study participants were selected by simple random sampling method in each Hospital. Details were presented in the main document.

Where did you get or how did you develop data collection tools or questionnaire in this study? Thanks editor for the comment. The questionnaire was adopted from Advanced corporate training (Time Management Questionnaire) and legal management consultants 2010 and put the citation in the main document.

How did you measure the validity and reliability of the study tools or questionnaire implemented to collect data in this study? 

 Thanks editor for the comment. To measure the validity of the study tools or questionnaires implemented to collect data in this study we have conducted pretest among 42 participants out of the study area (i.e. Addis-zemen hospital) and training was given for data collectors. Reliability test (Cronbach’s alpha) was performed to check reliability of the questionnaire items.

It is not also clear how the cutoff point made for the Likert scale

 Thanks for the comment. Cutoff points for independent variables were made based on their mean values as high (i.e. ≥ the mean score value) and low (i.e. below the mean value). But for the outcome variable i.e. time management practice was high/good if ≥ 65% and low if below 65% which was taken from journal of education and practice (Qteat M, 2014). 

How the questionnaire contents organized and formulate to address the objective?

 Thanks editor for the comment. In this study Questionnaire contents were organized by themes or main concepts from simple to complex for ease of respondents.

What is the dependent variable in this study? It looks like level of time management practice appears to be the dependent variable. Similarly, would be good to define those factors indicated in this study as well.

 Thank you editor for this constructive comment. The dependent variable of the study is time management practice not the level of time management practice and it was corrected and operationalized in the main document accordingly.

We have also defined factors in the main document.

Result

Maybe, it would be useful if you depict the response rate in percent than numbers for simplicity. Thanks for the comment. We have depicted the response rate as 92.65% in the main document.

It looks like there is a kind of mingling between the sub-topics “time management practices, organizational policies, employees’ performance and personal factors” and “factors associated with time management practice”. Either you need to modify the first sub topic or merge with the second topic.

 Thank you editor for the comment. We have modified the first subtopic in the main document and expressed it correctly as organizational policies, employees performance and personal factors of respondents. 

Discussion

It is not clear, for instance, on the first paragraph you indicated that “the proportion of time management practice among employees was 56.4%” what does the word proportion designate in this sentences? There is similar inconsistency throughout the document that has to be fixed. Thank you editor for the comment. We have changed proportion to prevalence and corrected as “the overall prevalence of time management practice among employees was 56.4%”. We have also made similar change throughout the document. 

In addition to this, you need to use studies that are closely related to your study setting at various measuring scale to make plausible comparison. Otherwise, citing studies which never have related to your study setting will affect the analysis and the entire result one or another way. Thank for the comment. Even though there were inadequate literatures we have included almost related studies to our setting in the main document.

What is the limitation of your study?

 Thank you editor for the comment. We have included the limitation of the study in main document.

Conclusion

You have inscribed your suggestion and recommendation at the abstract section, but not in this section Thanks again for the comment. We have included recommendations in the main document according to your query. 

Journal Requirements:

http://www.journals.plos.org/plosone/s/file?id=wjVg/PLOSOne_formatting_sample_main_body.pdf and 

Thank you for supporting us sharing the link to correct the manuscript format. Hence, the manuscript was corrected according to PLoS One format. 

2. Please include captions for your Supporting Information files at the end of your manuscript, and update any in-text citations to match accordingly. Please see our Supporting Information guidelines for moreinformation: http://journals.plos.org/plosone/s/supporting-information.

Thank you editor for supporting us sharing the link. We have included captions for the supportive information files at the end of the manuscript and updated any in-text citations. 

Reviewer #1

General comment

Is there an African / Ethiopia perspective of time? Thanks reviewer for your comment. 

Thus the African understands time as consisting of a long past and a present with virtually no future. This contrasts with the Western concept of time which is linear, consisting of an indefinite past, the present and infinite future. For the African, the future is absent since it has not been realized. African time or Africa time is the perceived cultural tendency, in parts of Africa and the Caribbean toward a more relaxed attitude to time. 

An African "emotional time consciousness" has been suggested which contrasts with Western "mechanical time consciousness". African cultures are often described as "polychronic", which means people tend to manage more than one thing at a time rather than in a strict sequence. Personal interactions and relationships are also managed in this way, such that it is not uncommon to have more than one simultaneous conversation.

The timezone in Ethiopia is EastAfricaTime (EAT) (UTC+03:00). Almost all Ethiopians use a 12-hour clock system. The daytime cycle begins at dawn 12:00 (6:00:00 AM EAT) and ends at dusk 11:59:59 (5:59:59 PM EAT). The night time cycle begins at dusk 12:00 (6:00:00 PM EAT) and ends at dawn 11:59:59 (5:59:59 AM EAT). The convention is that the day begins at 1:00 o'clock in the morning 12 hour cycle (7:00 AM EAT) rather than midnight (12:00 AM EAT). Therefore, the local population almost effectively observes UTC-03:00.

What are the dimensions of time? Thanks reviewer for your comments. Time is the dimension that allows things to do it, which is why we can measure the duration that they last or how fast they move. Space is where these things are and happen. Time and space are inextricably connected into what’s called the space-time continuum. The normal three dimensions including up-down, left-right, forward-back, and space-time. Two-dimensions of time would make time travel possible. Instead of being linear, at some point time loops back on itself. In this way, you could travel back or forward in time.

abstract Section

The objective is missed Thanks reviewer for your comments. We have included the objective in the main document. 

Avoid use of ‘predictors’ in cross sectional studies. Thank you reviewer for the comment and we have replaced predictors with factors associated with in the main document. 

In key words include Gondar Thanks again for the comment. We add Gondar in the key word in the abstract section of main document. 

Introduction

Is this definition of time the contemporary definition? Thanks reviewer for the comment. The definition of time was adopted from Mariam Webster dictionary and it is the contemporary definition used in this study.

What is special about time management for health care workers? Thanks again for the comment. 

Time management is about how one manages self. One cannot manage the time; one can only manage how he/she can use it. Organizing and prioritizing the patient care activities is of prime importance for providing quality care and to maintain the personal and professional balance. If healthcare workers spent their time out of serving their patient even within seconds they may lose lives and that is why time management for health care professionals is highly important priority issue.

What are the adverse consequences of ineffective time management practice?

 Thanks again for the comment. Some of the adverse consequences of ineffective time management practice are habitual lateness, overextension, inability to achieve goals, lack of success, and lack of confidence, stress, and ineffectiveness in one’s job as mentioned in many management books. 

The flow lacks coherence. Global, regional and national data on the matter of interest were missed. Thanks again for the comment. We have corrected the flow of ideas and incorporated global, national and regional data on time management practices in the main document.

Methods

Start with study setting and tell us your reference. Thanks reviewer for the comment. We have started with study setting and put the reference i.e. from North Gondar health department’s plan office data and corrected accordingly in the main document.

You didn’t mentioned study design. Thanks reviewer for the comment. We have mentioned the study design i.e. institutional based cross-sectional study design was used for this study in the main document.

Why you used systematic sampling? Thank you reviewer for this fruitful comment. Systematic sampling was inappropriate technique for this study and we have used appropriate sampling technique i.e. Simple random sampling and corrected accordingly in the main document. 

Is there no difference between male and female health care professionals? Between diploma and degree holders? Between nurses and medical doctors Thanks again for the comment. In this study the difference is their profession but regarding to our outcome variable every healthcare worker are expected to have almost the same time management practice for their own task because every task performance according to their time in the health care setting has a great place in the accomplishment of organizations goal.

Result

Why don’t you classify residence as urban and rural rather than presenting it as highlanders and lowlanders? Thanks reviewer for the comment. In this study residence is classified as highlanders and lowlanders but not as urban and rural because all primary hospitals are located in the surrounding rural districts of North Gondar zone. Rather primary hospitals are found in the highlands and lowlands that is why we classified residence as highlanders and lowlanders. Those employees who are working in hospitals found in the highland were classified as highlanders and those working in hospitals found in lowland were classified as lowlanders.

When do we say an individual is satisfied with performance appraisal?

 Thanks again for the comment. In this study when individuals are satisfied with performance appraisal if their response to the five items of five point likert scale questionnaires is above or equal to the mean score value of performance appraisals. 

What is your case to variable ratio?

 Thank again for the comment. In this study case to variable ratio is 0.565 (i.e. 221: 391 or 221/391). 

Present some of your findings e.g. Proportion of Procrastination Thanks for the comment. We have presented the findings clearly in the main document e.g. proportion of procrastination was 81.1% (i.e. 317 out of 391), planning was 80.8% (i.e. 316 out of 391) and also for others. 

Discussion

Avoid presenting frequencies. Thanks reviewer for the comment. We avoid frequencies from the main document and corrected it according to your inquiry. 

Reference for your justification is needed. For example “This might be due to unfair relationships between some workers and with the department..” what does that mean? Do you have data on it? Thanks reviewer for this golden comment. We have put reference for justifications in the main document. 

What does organizational policy? Is that norm or some other guiding document? Is that endorsed from ministry of health or?

 Thanks again for the comment. It is from the guiding document from ministry of health of Ethiopia that fosters positive working environment in the primary hospitals and the tools were prepared to assess the implementation of the policy and the working cultures of the hospitals as a factor for their time management practice. 

What are the types of compensations and benefits? Do you have evidence? Needs reference? Thanks for the comment. Types of compensations and benefits like financial, material, training, educational etc… are expected to be fulfilled for health care workers as to their educational level and performance. The evidence was from the human resource management for health guidelines of Ethiopian primary hospitals (EHRIG, 2014).

You collected data of 65% from diploma holders and you are concluding for all types of health care workers?

 Thanks reviewer for the comment. In Ethiopian context most of the primary hospitals are equipped with diploma holders staffs, hence degree and above degree holder staffs are low in number as compared to diploma holders, as our study showed from nine primary hospitals 65% of employees were diploma holders. This is according to the Ethiopian primary hospital standard guideline profession mix.

The word ‘poor’ is not ethical

 Thanks reviewer for the comment. We have used good or poor time management practice according to literature’s but now we have changed into high or low time management practice and corrected accordingly in the main document. 

Implications were missed Thanks again for the comment and we have included the implications of the study in the main document. The study could have valuable implication to formulate appropriate strategies by different stakeholders involved in hospital administration and management programs, and other related public health interventions. It has helped to enhance and upgrade participants’ understanding about time management; and to equip them with major techniques of time management. This study could also provide information to subsequent researchers on time management practices and factors associated with it.

Lacks recommendation

 Thanks again for the comment. We have added recommendation in the main document. Thus it is recommended that hospital employees must give special emphasis about planning to improve their time management practice. Hospital Managers need to assess and improve organizational policies and strategies and performance appraisal systems practicing in the respective hospitals to treat employees fairly and equally. It is recommended that Amara National Regional Health Bureau strengthen regular supportive supervision to the hospitals and time management training programs must be provided to hospital staffs in varies level at different health setting. It is recommended to conduct further research on the issue by extending the study setting and the study population.

---

## [Editor Report · Decision Letter 1]

15 Nov 2019

PONE-D-19-24146R1

Assessment of time management practices and associated factors among employees of primary hospitals in north Gondar, northwest Ethiopia

PLOS ONE

Dear Erkihun Tadesse,

Thank you for submitting your manuscript to PLOS ONE. After careful consideration, we feel that it has merit but does not fully meet PLOS ONE’s publication criteria as it currently stands. Therefore, we invite you to submit a revised version of the manuscript that addresses the points raised during the review process.

I do appreciated that you tend to incorporate the comments and feed backs given on the first submission. The second submission seems to be more organized and includes opinions relevant information to improve this manuscript. Being said that I have notice  few gaps that need to be address and bridge the gap and move forward to the  next phase of your manuscript.

The introduction in the abstract section need to be shorten and depict time management and purpose of the study in a clear fashion. In general, this section must show the entire study with limited words if possible with in one page.The introduction looks more organized and includes relevant literature that could be useful to make good argument.Methods:- I haven't seen a single statement that show the operational definition of your study variable. It would be also very useful to include the dependent and independent variable and its operational definition to provide a clear picture of this study.The result section appears to be improved at a different perspective, however, I recommend you to separate the subtopic organization policy, employment performance and personal factors. Perhaps, that would help you to show the detail analysis of this study. The discussion well written and improved significantly. However, still you need to keep the flow of the write up based on the analysis on the result section, which will help you maintain  the coherence of statement in alignment with the result section. 

We would appreciate receiving your revised manuscript by Dec 30 2019 11:59PM. To enhance the reproducibility of your results, we recommend that if applicable you deposit your laboratory protocols in protocols.io, where a protocol can be assigned its own identifier (DOI) such that it can be cited independently in the future. For instructions see: http://journals.plos.org/plosone/s/submission-guidelines#loc-laboratory-protocols

We look forward to receiving your revised manuscript.

Kind regards,

Solomon Assefa Woreta

Academic Editor

PLOS ONE

Additional Editor Comments (if provided):

I do appreciated that you tend to incorporate the comments and feed backs given on the first submission. The second submission more organized and include opinions relevant to improve this manuscript. Being said that I have notice few gaps that need to be address to move the next phase of your manuscript.

- The introduction in the abstract section need to be shorten and show time management and purpose of the study. Overall, this section should briefly depict the entire study as least in one page.

-The introduction looks more organized and includes relevant literature that could be useful to make good argument.

-Methods:- I haven't seen a single statement that show the operational definition of the study variable in this study. Would be also very useful to include the dependent and independent variable and its operational definition to provide a clear picture of this study.

-The result section appears to be improved at a different perspective, however, I recommend you to separate the subtopic organization policy, employment performance and personal factors. Perhaps, that would help you to show the detail analysis of this study.

-The discussion well written and improved significantly. However, still you need to keep the flow of the write up based on the analysis on the result section, which will help you maintain the coherence of statement in alignment with the result section.

---

## [Author Response · Author response to Decision Letter 1]

29 Nov 2019

We would like to thanks for these constructive, building and improvable comments on this manuscript that would improve substance and content of the manuscript. We considered each comments and clarification questions of editors and reviewers on the manuscript thoroughly. Our point-by-point responses for each comment and questions are described in detailed on the following pages. Further, the details of changes were shown by track changes in the supplementary document attached.

Editor questions/comments Response

Abstract

The introduction in the abstract section need to be shorten and depict time management and purpose of the study in a clear fashion. In general, this section must show the entire study with limited words if possible with in one page.

 Thanks editor for your constructive comment. We have corrected this section by depicting time management practice and purpose of the stud in the main document according to your query. 

Methodology

I haven't seen a single statement that shows the operational definition of your study variable. It would be also very useful to include the dependent and independent variable and its operational definition to provide a clear picture of this study. Dear editor thanks for your constructive comments. We have included the dependent and independent variable, and its operational definition in the main document as shown in the truck number from 146 – 192. 

Result

The Result section appears to be improved at a different perspective, however, I recommend you to separate the subtopic organization policy, employment performance and personal factors. 

 Thank you editor for the comment. We have separated the organization related factors of respondents, Employees performance related factors and personal related factors of respondents separately in the main document. 

Discussion

The discussion well written and improved significantly. However, still you need to keep the flow of the write up based on the analysis on the result section, which will help you maintain the coherence of statement in alignment with the result section. 

 Thank you editor for the comment. We have arranged the discussion section according to the result section in the main document.

---

## [Editor Report · Decision Letter 2]

27 Dec 2019

PONE-D-19-24146R2

Assessment of time management practice and associated factors among primary hospitals employees of in north Gondar, northwest Ethiopia

PLOS ONE

Dear Dr Amsalu,

Thank you for submitting your manuscript to PLOS ONE. After careful consideration, we feel that it has merit but does not fully meet PLOS ONE’s publication criteria as it currently stands. Therefore, we invite you to submit a revised version of the manuscript that addresses the points raised during the review process.

We would appreciate receiving your revised manuscript by Feb 10 2020 11:59PM. To enhance the reproducibility of your results, we recommend that if applicable you deposit your laboratory protocols in protocols.io, where a protocol can be assigned its own identifier (DOI) such that it can be cited independently in the future. For instructions see: http://journals.plos.org/plosone/s/submission-guidelines#loc-laboratory-protocols

We look forward to receiving your revised manuscript.

Kind regards,

Solomon Assefa Woreta

Academic Editor

PLOS ONE

Journal Requirements:

Additional Editor Comments (if provided):

I would like again to commend authors for the prompt work to improve the manuscript. The manuscripts appears to be further improved from the original submission.

Abstract:

Briefly illustrate the overall of the study and provides precise information.

Introduction:

I have noted the significant change made on the background section. I advise you to remove the last statement after the purpose of the study that appears to be a recommendation and the quote to refer previous study, you don’t have a tangible assertion whether the same topic conducted or not in Ethiopia setting. Please remove the following paragraph “which was not studied previously. Thus the findings from the study would provide valid information for policy makers, managers and stakeholders for taking appropriate action to improve time management practice.”

Methods:

This section seems well written and have significant improvement. It composed of necessary tools to undertake the study.

Result:

It seems to have included the comments provided by the reviewers. The statistical analysis sound and genuine.

Discussion:

The big gap I have noticed in this section is most of the arguments have never been supported by study or evidence. Your arguments need to have reasonable evidence. Your assumption to elucidate the discrepancy originally inferred based on what you thought about it, which is not scientifically sound even if your arguments are correct. There is plenty of research out there that would support your assumption, hence I recommend you to rewrite this section using references that bolster your assertion.

Conclusion:

Make sure your conclusion and recommendation based on merely on the finding of this study. Don’t try to additional points out this study context.
---

## [Author Response · Author response to Decision Letter 2]

31 Dec 2019

Rebuttal letter Date: January/01/2020

PONE-D-19-24146

Assessment of time management practice and associated factors among primary hospitals employees in north Gondar, northwest Ethiopia 

Erkihun Tadesse

PLOS ONE

Dear all,

We would like to thanks for these constructive, building and improvable comments on this manuscript that would improve substance and content of the manuscript. We considered each comments and clarification questions of editors and reviewers on the manuscript thoroughly. Our point-by-point responses for each comment and questions are described in detailed on the following pages. Further, the details of changes were shown by track changes in the supplementary document attached.

Editor questions/comments Response

Introduction

I advise you to remove the last statement after the purpose of the study that appears to be a recommendation and the quote to refer previous study, you don’t have a tangible assertion whether the same topic conducted or not in Ethiopia setting. 

 Please remove the following paragraph “which was not studied previously. Thus the findings from the study would provide valid information for policy makers, managers and stakeholders for taking appropriate action to improve time management practice.” Thanks editor for your constructive comment. We have removed the paragraph “which was not studied previously. Thus the findings from the study would provide valid information for policy makers, managers and stakeholders for taking appropriate action to improve time management practice.” Expressed as Therefore, this study aimed to assess time management practices and associated factors among employees of primary hospitals in North Gondar Zone. 

Discussion

Your arguments need to have reasonable evidence. Your assumption to elucidate the discrepancy originally inferred based on what you thought about it, which is not scientifically sound even if your arguments are correct. There is plenty of research out there that would support your assumption, hence I recommend you to rewrite this section using references that bolster your assertion. Thank you editor for the comment. We have supported the arguments/assumptions using references and corrected according to your inquiry in the main documents. 

Conclusion

Make sure your conclusion and recommendation based on merely on the finding of this study. Don’t try to additional points out this study context.

 Thank you editor for constructive comments. We have presented the conclusion and recommendation based on the finding of this study and corrected according to your inquiry.

---

## [Editor Report · Decision Letter 3]

6 Jan 2020

Assessment of time management practice and associated factors among primary hospitals employees in north Gondar, northwest Ethiopia

PONE-D-19-24146R3

Dear Dr. Erkihun Tadesse,

We are pleased to inform you that your manuscript has been judged scientifically suitable for publication and will be formally accepted for publication once it complies with all outstanding technical requirements.

With kind regards,

Solomon Assefa Woreta

Academic Editor

PLOS ONE
---

## [Editor Report · Acceptance letter]

10 Jan 2020

PONE-D-19-24146R3 

Assessment of time management practice and associated factors among primary hospitals employees in north Gondar, northwest Ethiopia 

Dear Dr. Amsalu:

I am pleased to inform you that your manuscript has been deemed suitable for publication in PLOS ONE. Congratulations! Your manuscript is now with our production department. 

With kind regards,

on behalf of

Dr. Solomon Assefa Woreta 

Academic Editor

PLOS ONE